# Neuroendocrine Assessment of Dopaminergic Function during Antidepressant Treatment in Major Depressed Patients

**DOI:** 10.3390/brainsci11040425

**Published:** 2021-03-26

**Authors:** Fabrice Duval, Marie-Claude Mokrani, Alexis Erb, Felix Gonzalez Lopera, Vlad Danila, Mihaela Tomsa

**Affiliations:** Pôle 8/9-APF2R, Centre Hospitalier, 68250 Rouffach, France; marieclaudejean2010@hotmail.fr (M.-C.M.); a.erb@ch-rouffach.fr (A.E.); f.gonzalez@ch-rouffach.fr (F.G.L.); v.danila@ch-rouffach.fr (V.D.); m.pop@ch-rouffach.fr (M.T.)

**Keywords:** dopamine, antidepressant, apomorphine, cortisol, dexamethasone, remission

## Abstract

The effects of antidepressants on dopamine (DA) receptor sensitivity in the mesolimbic–hypothalamic system have yielded contradictory results. The postsynaptic DA receptor function was evaluated by the cortisol response to apomorphine (APO; 0.75 mg SC) in 16 drug-free DSM-5 major depressed inpatients and 18 healthy hospitalized control (HC) subjects. Cortisol response to the dexamethasone suppression test (DST) was also measured. After two and four weeks of antidepressant treatment (ADT), the DST and APO test were repeated in all patients. Cortisol response to APO (∆COR) was not influenced by the hypothalamic–pituitary–adrenal (HPA) axis activity, as assessed by the DST. Pre-treatment ∆COR values did not differ significantly between patients and HCs. During ADT, ∆COR values were lower than in HCs at week 2 and 4. After four weeks of treatment, among the eight patients who had blunted ∆COR values, seven were subsequent remitters, while among the eight patients who had normal ∆COR values, seven were non-remitters. Considering the limitations of our study, the results suggest that following chronic ADT, the desensitization of postsynaptic DA receptors connected with the regulation of the HPA axis at the hypothalamic level is associated with clinical remission. These results could reflect increased DA levels in the mesolimbic pathway.

## 1. Introduction

Accumulating evidence suggests that hypofunction of the mesolimbic dopamine (DA) pathway is involved in major depression, particularly in anhedonia and loss of motivation [1,2]. Since pioneering studies reporting that concentrations of homovanillic acid (a DA metabolite) were decreased in the cerebrospinal fluid of depressed patients, several neuroimaging and binding studies, but not all [3,4], have revealed increased binding of D_2_-like receptors in the mesocorticolimbic DA system, possibly secondary to decreased presynaptic DA release [5,6,7,8]. In addition, pharmacological studies have reported that DA receptor agonists (e.g., bromocriptine, pergolide, piribedil, pramipexole) exhibit some antidepressant properties [9,10,11], while DA receptor antagonists may worsen depressive symptoms [12].

Several preclinical studies have shown that chronic treatment with different classes of antidepressant drugs may increase the functional responsiveness and binding of DA-D_2/3_ receptors in the mesolimbic system [13,14,15,16,17,18]. Thus, it has been hypothesized that sensitization of D_2/3_ receptors may represent a “final common pathway” in antidepressant action [19]. However, pharmaco-neuroimaging studies have reported conflicting data on the DAergic effects of various antidepressants in depressed patients [3,20,21,22,23,24]. Hence, to deepen our understanding of the possible role of DA changes in the therapeutic response to antidepressants, functional investigations focusing on areas of the brain other than those usually studied (e.g., striatum, basal ganglia, frontal cortex, or thalamus) are needed. In this context, dynamic investigations of the hypothalamic–pituitary system—which is the primary output node for the limbic system—using DAergic agents appear relevant [25].

In psychiatric patients, central DAergic neurotransmission may be indirectly assessed by means of the hormonal responses to the non-selective DA receptor agonist apomorphine (APO), administered subcutaneously (SC) [26]. APO inhibits prolactin (PRL) secretion (via D_2_ receptors of pituitary lactotrophs) and stimulates growth hormone (GH) and hormones of the hypothalamic–pituitary–adrenal (HPA) axis, i.e., adrenocorticotropic hormone (ACTH) and cortisol (COR), via the agonization of DA receptors on neurons in the hypothalamus which regulate the release of growth hormone-releasing hormone (GHRH) and corticotropin-releasing hormone (CRH) [27]. Both in depressed patients and healthy controls GH, ACTH, and COR—but not PRL—responses to APO are interrelated, suggesting common pathways at the hypothalamic level [27,28]. However, studies of hormonal responses to APO in depressed patients vs. controls are inconclusive: increased, decreased, or unchanged responses have been reported [27,28,29,30,31,32,33,34,35,36,37]. These discrepancies could be explained by methodological differences between studies (in particular regarding the dose of APO administered; 0.5 mg or 0.75 mg SC), but also by the bio-clinical heterogeneity of depressed patients. Studies on the therapeutic applications of the APO test are few and inconsistent: in a small and nonreplicated study, blunted GH response to APO (0.5 mg SC) in depressed patients at baseline (i.e., after a two-week drug-free period) predicted a favorable response to paroxetine [38]; in a study evaluating six healthy male volunteers, repeated amitriptyline administration decreased GH response to APO [39]; in another study, it has been found that two and six weeks of treatment with a range of antidepressant drugs and/or ECT did not affect the GH or PRL responses to APO in depressed patients [40]. Until now, the effect of antidepressant drugs on the COR response to APO has not been studied in depression.

The aim of this study was to evaluate the changes in DAergic activity following two and four weeks of antidepressant treatment (ADT) using the COR response to APO as a probe of DAergic receptor responsiveness. Concomitantly, we performed a dexamethasone suppression test (DST) to assess the potential impact of the HPA axis activity on the COR response to APO.

## 2. Materials and Methods

### 2.1. Subjects

This was a retrospective study carried out on patient files. The data were extracted from a bio-clinical database based on a series of neuroendocrine challenge tests. This multisystem approach was approved by the local ethics committee (Rouffach Hospital Review Board) and carried out in accordance with the Declaration of Helsinki. Sixteen DSM-IV major depressed hospitalized patients [41] (6 men and 10 women, mean age ± standard deviation, 38.7 ± 10.5 years) without psychotic characteristics and having completed at least 6 weeks of ADT were enrolled in this study. Patients were recruited between 1999 and 2011 from the inpatient units of the Pole 8/9, Psychiatric Hospital of Rouffach (France), and were diagnosed by an experienced psychiatrist blind to endocrine results by means of an unstructured clinical interview. To participate in this study, patients needed to be adults (≥18 years of age) and to have a baseline score ≥18 on the 17-item Hamilton Depression Rating Scale (HAM-D]) [41] (mean ± SD score, 26.8 ± 4.6). All enrolled patients were treated with antidepressants at the time of hospital admission, and all were free of psychotropic drugs for at least 2 weeks before the baseline neuroendocrine tests (the washout period was supervised in hospital). All subjects gave their informed consent prior to participation. To evaluate the level of DAergic and HPA axis activity in patients, we selected data of 18 hospitalized healthy volunteers (9 men and 9 women, mean age ± SD, 37.8 ± 7.8 years) from our database. None of them had a personal or family history of major psychiatric illness, or met criteria for Axis I diagnostic, or had been previously treated with psychotropic medications.

To exclude concomitant medical conditions, all participants underwent a medical examination and blood testing. Beyond routine dosages, thyroid axis hormones (thyrotropin (TSH), free thyroxine, free triiodothyronine), HPA axis hormones (ACTH, COR), and PRL and GH values had to be within the normal range. Participants’ body mass indexes needed to be in the normal range, i.e., 18.5–25 kg/m^2^. Exclusion criteria were a personal history of endocrine disease; drug abuse (including alcoholism but excluding nicotine); and past treatment with monoamine oxidase inhibitors, lithium salts, carbamazepine, long-acting antipsychotics, fluoxetine (or other antidepressant with a very long half-life of elimination), and/or electroconvulsive therapy. All women were free from hormonal contraception and were tested outside the periovulatory period. Patients and controls were on a caffeine-free diet at least three days prior to the biological assessment; their nyctohemeral rhythm was synchronized with diurnal activity from 8:00 a.m. to 11:00 p.m.

### 2.2. Procedures

A DST was performed with the oral ingestion of 1 mg of dexamethasone (Dectancyl, Laboratoires Roussel, Paris, France) at midnight, and serum COR levels were measured at 8:00 a.m., 4:00 p.m., and 11:00 p.m. on the next day. The APO test was performed three days after the DST [35]. After an overnight fasting, subjects were woken at 7:00 a.m., and a cannula was inserted into an antecubital vein of the arm. Baseline COR levels were assessed at −30, −15, and 0 min. Then, a sub-emetic dose of APO (0.75 mg SC, Apokinon, Laboratoires Aguettant, France) was administered at 9:00 a.m. Blood samples were collected at +15, +30, +60, + 90, +120, and +150 min. Throughout the test, subjects remained in bed and did not smoke. After 2 and 4 weeks of inpatient ADT, the same neuroendocrine tests were repeated in all patients (i.e., on Day 14 and Day 28).

### 2.3. Assays

Blood samples were immediately stored at 4 °C until their use for assays. COR concentrations were determined by immunometric assay based on enhanced luminescence. Average intra-assay and inter-assay coefficients of variation were 6.2–8.9%, respectively, sensitivity <3 nmol/L (Amerlite Cortisol Assay^®^, Amersham, Amersham, UK).

### 2.4. Antidepressant Treatment

After the washout period, followed by the pre-treatment neuroendocrine assessment, antidepressants were given under supervision on the ward, at doses known to be effective [42]. The choice of treatment, which was not an object of the study, was made on a clinical basis [42]. At the time of the first neuroendocrine retest, on Day 14, 8 patients received venlafaxine extended release (VFX, a serotonin (5-HT)–norepinephrine (NA) reuptake inhibitor; mean dose ± SD, 154.7 ± 31.3 mg/d; range 112.5–225 mg/d), and 8 received tianeptine (TIA, a 5-HT reuptake enhancer and a combined µ- and δ-opioid receptor agonist) [43]; TIA began at 37.5 mg/d then was raised to 75 mg/d after one week; this dose was maintained stable throughout the treatment period. Regarding VFX, changes in dose were made, if necessary, after 2 weeks, according to clinical response and tolerance; mean doses were on Day 28, 187.5 ± 56.7 mg/d (range 150–375 mg/d), and on Day 42, 215.6 ± 93.5 (range 150–375 mg/d). Psychotropic drugs other than the study medication were, if possible, avoided. Alimemazine (maximum dose: 30 mg/d orally) was allowed in cases of severe sleep disturbance. Patients were hospitalized for a minimum of 4 weeks; 3 patients could be discharged from hospital before reaching Day 42 and were followed as outpatients until the end of the study, whereas the other patients remained hospitalized throughout the study (6 weeks).

### 2.5. Clinical Response

All patients were assessed weekly using the 17-item HAM-D scale—raters being blind to the DST and APO test results and the medication used. Patients were not aware of the neuroendocrine results. Clinical remission was defined as a final HAM-D score (on Day 42) less than 8 [44].

### 2.6. Data Analysis

Given the marked decrease in COR concentration in the morning—due to the normal circadian rhythm—we used the COR value at 0 min (i.e., immediately before injection of APO) as the baseline value (COR*_BL_*); COR response was measured by subtracting the baseline level from the peak level after APO (∆COR) [27]. To evaluate the COR suppression to dexamethasone administration we used the maximum COR (COR*_max_*) level in any blood sample obtained at 8:00 a.m., 4:00 p.m., and 11:00 p.m. [45].

Statistical analyses were performed using software from the R Project for Statistical Computing [46]. We employed nonparametric statistical tests because some data were not normally distributed (according to the Kolmogorov–Smirnov test). Comparisons between control subjects and depressed patients were tested with the Mann–Whitney two-tailed test (U test)—formal corrections for multiple comparisons were not needed because we made planned comparisons. Within-patient differences were evaluated with the Friedman rank sum test and, where the overall effect was significant, by the Wilcoxon two-tailed signed-rank test (*t*-test) for paired data. Relationships between quantitative data were estimated with the Spearman’s rho (ρ) statistic. Optimal cut-off values were determined using receiver operating characteristic (ROC) curve analysis [47]; qualitative data were analyzed using the Fisher’s exact test (two-tailed). Logistic regression was used to estimate the probability of remission/non-remission on Day 42 based on ∆COR values after 2 and 4 weeks of treatment. Results were considered significant when *p* ≤ 0.05.

## 3. Results

### 3.1. Comparison between Depressed Patients and Control Subjects

Patients and controls were comparable for age (*p* = 0.7 by U test) and sex (*p* = 0.5 by Fisher’s exact test). Overall, COR values (i.e., COR*_BL_*, ∆COR, and post-DST COR*_max_*) were not significantly different between untreated patients and controls (Figure 1), and were not influenced by age, gender, or severity of depression at baseline (Day 0), as evaluated with the HAM-D scores. COR values did not statistically change during ADT (COR*_BL_*, *p* = 0.77; ∆COR, *p* = 0.13; post-DST COR*_max_*, *p* = 0.42, by Friedman test). Nonetheless, ∆COR values were significantly lower in patients than in controls on Day 14 and on Day 28. No meaningful relationships were found between ∆COR and COR*_BL_*, or between ∆COR and post-DST COR*_max_* values at baseline, on Day 14, or on Day 28.

As illustrated in Figure 2, when using a ∆COR value of less than 0 nmol/L to define a blunted response, four untreated patients (25%) and one control (5.5%) showed a blunted response (*p* = 0.15 by Fisher’s exact test). Five patients (31%) on Day 14, and eight on Day 28 (50%) showed blunted ∆COR values (*p* = 0.07, and *p* = 0.005, respectively, vs. controls).

Although ∆COR values were not statistically altered during ADT for the depressed group as a whole, there were, however, noticeable changes at the individual level (Figure 2). The extent of ∆COR changes between Day 0 and Day 14 (i.e., ∆∆COR_Day14–Day0_) was negatively related to pre-treatment ∆COR values (ρ = −0.78; *n* = 16, *p* < 0.0006). Such a negative correlation was also found between ∆COR values on Day 14 and their evolution between Day 14 and Day 28 (∆∆COR_Day28–Day14_) (ρ = −0.71; *n* = 16, *p* < 0.002).

Regarding treatment groups, COR values at baseline and during treatment were comparable between VFX and TIA groups (Table 1). APO-induced COR stimulation was not changed by either compound (VFX group, *p* = 0.51; TIA group, *p* = 0.20 by Friedman test).

### 3.2. Apomorphine-Induced Cortisol Stimulation and Clinical Outcome

As shown in Table 1, the clinical efficacy of VFX and TIA was comparable. In the whole depressed group, ∆COR values and HAM-D scores were unrelated on Day 14 (ρ = −0.26; *n* = 18; *p* = 0.34). However, ∆COR values on Day 28 were correlated with HAM-D scores on Days 28 and 42 (ρ = 0.62 and 0.67, respectively; *n* = 18; both *p* ≤ 0.01). Patients who showed blunted ∆COR values following four weeks of ADT were more likely subsequent remitters: among the eight patients who had blunted ∆COR values on Day 28, all but one were remitters; conversely, seven of the eight patients with normal ∆COR values were non-remitters (*p* = 0.01 by Fisher’s exact test) (Figure 2).

Although pre-treatment COR values and values on Day 14 did not distinguish subsequent remitters and non-remitters, ∆COR values on Day 28 were significantly lower in remitters than in non-remitters (Table 2). Compared to controls, ∆COR values in remitters were slightly lower at baseline (*p* = 0.06 by U test), normal on Day 14 (*p* = 0.12 by U test), and greatly reduced on Day 28 (*p* = 0.0002 by U test). In non-remitters, ∆COR values were comparable to those of controls at baseline (*p* = 0.27 by U test), blunted on Day 14 (*p* = 0.01 by U test), but were no longer significantly diminished on Day 28 (*p* = 0.06 by U test). Thus, the evolution of ∆COR value profiles during treatment was not superimposable between remitters and non-remitters: ∆COR values decreased between Day 0 and Day 14 in non-remitters (*p* = 0.05 by *t*-test) but not in remitters, while ∆COR values decreased between Day 14 and Day 28 in remitters only (*p* = 0.007 by *t*-test). The magnitude of the ∆COR decrement between Day 14 and Day 28 (∆∆COR_Day28–Day14_) was greater in remitters than in non-remitters (Table 2). ∆∆COR_Day28–Day14_ values were significantly negatively correlated with ∆COR values on Day 14 in remitters (ρ = −0.95; *n* = 8; *p* = 0.001), but not in non-remitters (*p* = 0.13). Among the eight patients who showed ∆∆COR_Day28–Day14_ less than −20 nmol/L, almost 88% (7/8) were remitters, while seven of the eight patients having COR_Day28–Day14_ values greater than −20 nmol/L were non-remitters (*p* = 0.01 by Fisher’s exact test). Figure 3 displays models provided by simple logistic regression analysis, predicting the likelihood of remission for any value of ∆COR on Day 28 (Figure 3A) and ∆∆COR_Day28–Day14_ (Figure 3B).

## 4. Discussion

The most striking finding from this present study is that patients who showed blunted ∆COR values after four weeks of ADT were more often subsequent remitters than those with normal values. Thus, our results suggest that following chronic antidepressant treatment, regardless of the compound administered (TIA or VFX), desensitization of postsynaptic DA receptors connected with the regulation of the HPA axis at the hypothalamic level is associated with clinical remission.

APO is a non-selective DA agonist which activates D_1_-like (D_1_, D_5_) and D_2_-like (D_2S/2L_, D_3_, D_4_) receptors [48,49]. Confirming our previous studies [27,28,37,50,51], APO-induced COR stimulation appears to be independent of basal HPA activity, as reflected by COR*_BL_*, and of glucocorticoid receptor sensitivity, as reflected by post-DST COR*_max_* values. Given that the COR response to APO is strongly correlated with that of ACTH [27,28,35,37,50,51], the latter being triggered by CRH via the stimulation of D_1_-like and D_2_-like receptors [52,53,54], ∆COR may indirectly assess the functionality of hypothalamic DA D_1_- and D_2_-like receptors. Interestingly, the functionality of pituitary D_2_ receptors—which inhibit the secretion of PRL and TSH, and probably also ACTH—is not impaired in depressed patients [27,28,35]. Consistent with previous reports from our group [27,37,50,51], ∆COR values were not significantly different between untreated patients and control subjects, suggesting that central DA D_1_- and/or D_2_-like receptors were functionally unaltered in the depressed group considered as a whole. Nevertheless, 25% of untreated patient exhibited blunted APO-induced COR stimulation, which may reflect a decrease in DA receptor responsiveness related to a reduction in tonic DA tone [2,28].

Following two and four weeks of ADT, APO-induced COR responses were weaker in patients than in controls. This finding may suggest that hypothalamic DA receptors connected with the regulation of the HPA axis are desensitized in some patients. As evidenced by the correlations between ∆COR and ∆∆COR values (after two and four weeks of treatment), the extent of desensitization of postsynaptic DA receptors over time of antidepressant exposure was related to the DA receptor sensitivity status (at baseline and after two weeks of treatment). Although animal studies have shown that most antidepressants (irrespective of their primary mechanism of action) have pro-DAergic effects by increasing the spontaneous activity of DA neurons in the mesolimbic system [55,56], imaging studies in depressed patients are inconclusive [3,20,21,22,23,24]. Unlike VFX, which increases the synaptic availability of 5-HT and NA by blocking the 5-HT and NA transporters, TIA does not bind to aminergic receptors or transporters and does not affect the extracellular levels of 5-HT, NA and DA; instead, the effects of TIA have been linked to changes in the glutamatergic system [57]. In our study, VFX and TIA, either administered subchronically or chronically, did not significantly change ∆COR values, suggesting that these compounds have no direct or indirect effect on hypothalamic D_2_- and/or D_1_-like receptor sensitivity. Therefore, blunted ∆COR values following four weeks of treatment seem related to successful treatment response rather than to a direct pharmacological effect of the drugs. The findings of Ebert et al. [3] also point in this direction, because decreased striatal D_2_ receptor binding was associated with a good clinical response to tricyclic antidepressants, while D_2_ receptor binding remained unchanged in non-responders.

In our study, the remission rate was slightly higher than that usually found in the literature [58], although it was in accordance with previous studies showing that two-thirds of patients ultimately achieved remission [59]. Thus far, individual differences in response to ADT are poorly understood [60], but a major determinant could be the patient’s “biological state” [61]. Although pre-treatment ∆COR values are slightly reduced in subsequent remitters vs. controls, they do not distinguish remitters and non-remitters. Thus, we cannot confirm the hypothesis of Healy and McKeon [38] that patients with DA receptor hyposensitivity at baseline would be most likely to have a favorable antidepressant response. Furthermore, the relationship between clinical outcome and DA receptor sensitivity during ADT remains a matter of debate, because sensitization and desensitization of these receptors have been both reported during ADT [3,18,62]. Preclinical data show that prolonged increase in DA concentration within the mesocorticolimbic system leads to DA receptor desensitization [63,64]. In the mesolimbic system, desensitization of D_2_ autoreceptors further increases DA output [63,65,66]. In our study, DA receptor desensitization was particularly observable between two and four weeks of treatment in subsequent remitters, as evidenced by the robust decrement of ∆COR values between Day 14 and Day 28 (i.e., ∆∆COR_Day28–Day14_). Moreover, the extent of desensitization following four weeks of treatment appeared linked to the state of DA receptors on Day 14. It is therefore conceivable that desensitization of D_1_- and/or D_2_-like receptors at the hypothalamic level would be adaptive to a long-term increase in DA output in the mesolimbic–hypothalamic pathway in remitters. Therefore, an intriguing hypothesis would be that the clinical effectiveness of the antidepressants depends on their capacity to stimulate the mesolimbic DA system activity.

In non-remitters, the transient decrease in ∆COR values during the first two weeks of ADT suggests early desensitization of hypothalamic DA receptors followed by a resensitization of these receptors. As a result, the reversible desensitization/resensitization of autoinhibitory D_2_ receptors at the presynaptic level could fail to sustainably increase DA tone in the mesolimbic–hypothalamic pathway. However, little is known about the mechanisms underlying the reversible desensitization/resensitization of DA receptors [67,68], particularly during ADT. One may hypothesize that this phenomenon could impede the positive outcome and promote the development of treatment-resistant depression. The effective approach of adjunctive treatment with DA agonists (e.g., pramipexole, ropinirole, bromocriptine) for pharmaco-resistant patients supports this view [9,10,11]. Further studies are required to confirm this hypothesis.

Despite the strengths of this study (i.e., each inpatient was tested at baseline—after a sufficient washout period to avoid bias due to a remanent effect of previous medications [27]—and retested after two and four weeks of ADT using the same standardized neuroendocrine assessment; COR values in patients were compared with those of healthy control subjects; the antidepressant response was evaluated after 6 weeks of treatment), some limitations relating to open-label small studies should be considered. First, the modest sample size may limit the generalizability of our results. Secondly, type I and type II errors may have been made. However, there was consistency between COR responses to APO (i.e., ∆COR and ∆∆COR) and clinical outcome (regardless of the statistical tests used to analyze data), which suggests some validity in our results. Thirdly, although the raters were blind to the medication used as well as the COR results, treatment was given in an open-label manner. We used two classes of compounds in order to optimize the ADT response (by considering the side effects of each drug and the previous antidepressant response, because all patients were treated at the time of their admission with inconsistent results). Thus, we cannot conclude whether change in DAergic transmission was involved in the antidepressant response in general, or if it was limited to specific antidepressant classes. Additionally, given the small sample size, it was not possible to statistically analyze the endocrine effects of each drug in relation to its clinical efficacy. For the same reason, failure to find a statistically significant difference must be interpreted with caution, because the power of the study may have been too low to detect such difference. Moreover, we did not use placebo as a comparative treatment because the aim of our study was not to demonstrate the effectiveness of antidepressant drugs, but to assess the endocrine correlates of ADT. Furthermore, we did not measure serum dexamethasone and APO; thus, we cannot rule out possible biases due to pharmacokinetic differences. However, as previously discussed, dexamethasone and APO serum levels are unlikely to account for differences in the results in depressed patients [28]. Finally, APO has also affinity for serotonin receptors (5-HT_1A_, 5-HT_2A_, 5-HT_2B_, and 5-HT_2C_), and α-adrenergic receptors (α_1B_, α_1D_, α_2A_, α_2B_, and α_2C_) [48,49]. Most of them have been involved, to different degrees, in the regulation of CRH activity [69]. Consequently, the changes in APO-induced COR stimulation might also partly reflect 5-HT and α-adrenergic receptor sensitivity, although this hypothesis needs further investigation in depressed patients.

## 5. Conclusions

To summarize, our study provides some evidence that the APO-COR test status after four weeks of ADT—unlike that at baseline or after two weeks of treatment—is associated with the clinical outcome. These findings suggest that following chronic ADT, desensitization of postsynaptic DA receptors connected with the regulation of the HPA axis at the hypothalamic level (possibly adaptive to increased DA output in the mesolimbic–hypothalamic system) is associated with clinical remission. Our findings, which provide new insights into the possible role of DA in the response to treatment, should be considered preliminary until replicated in a larger population.

## Figures and Tables

**Figure 1 brainsci-11-00425-f001:**
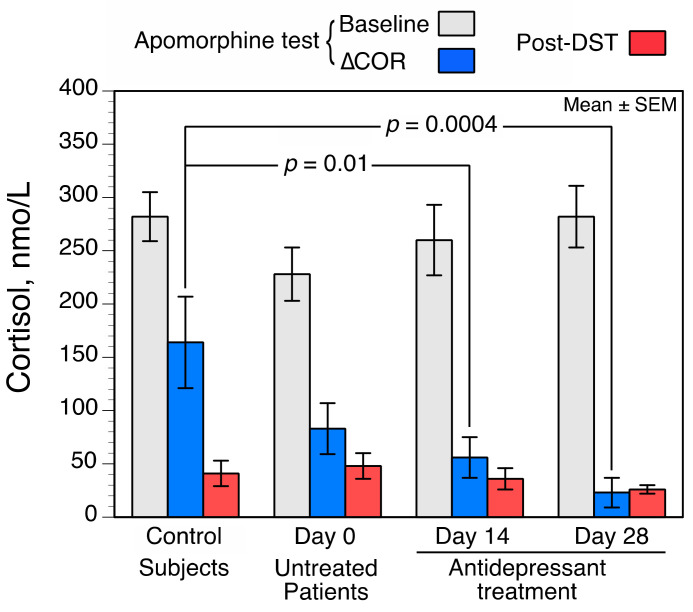
Cortisol values before (i.e., baseline, COR*_BL_*) and after 0.75 mg subcutaneously (SC) of apomorphine (i.e., ∆COR), and dexamethasone (i.e., post- dexamethasone suppression test (DST) COR*_max_*) in controls (*n* = 18) and in depressed patients (*n* = 16). Comparison between controls and patients was by Mann–Whitney two-tailed U test.

**Figure 2 brainsci-11-00425-f002:**
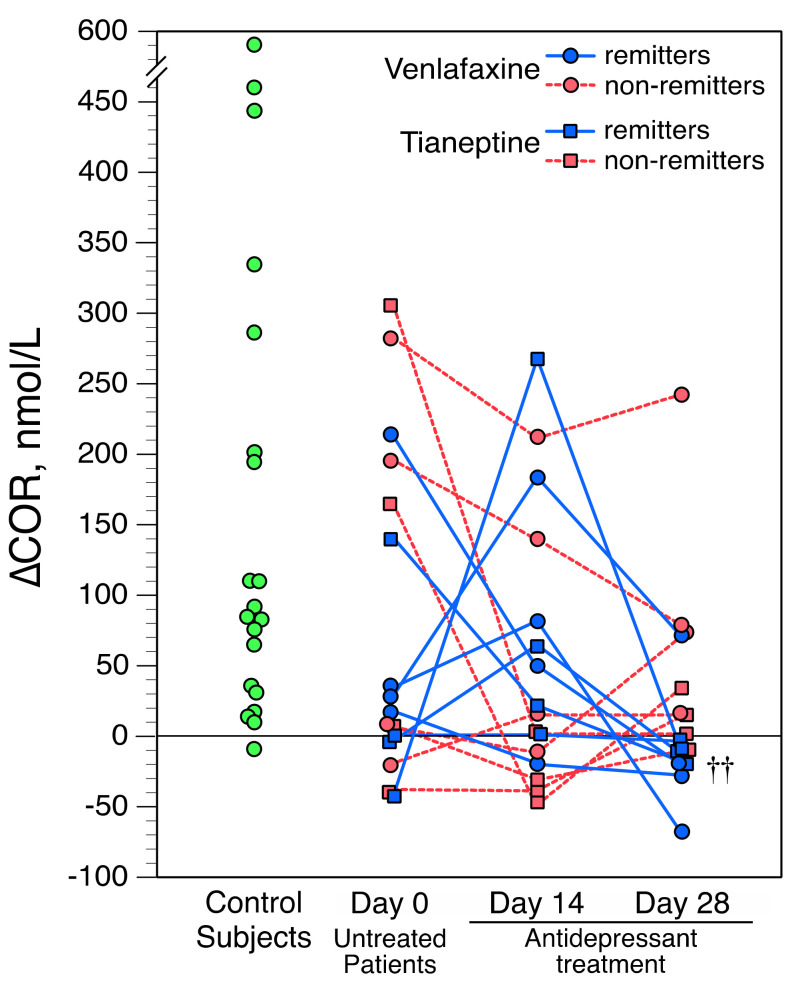
Apomorphine-induced cortisol stimulation (i.e., ∆COR) in controls and in depressed patients. Values are plotted individually. Regarding patients, circles represent those treated with venlafaxine; squares represent those treated with tianeptine; subsequent remitters (after 6 weeks of treatment) are marked in blue, non-remitters are in red. Threshold for a blunted ∆COR value, <0 nmol/L. Comparison between remitters and non-remitters on Day 28, †† *p* < 0.01 (by Mann–Whitney two-tailed U test).

**Figure 3 brainsci-11-00425-f003:**
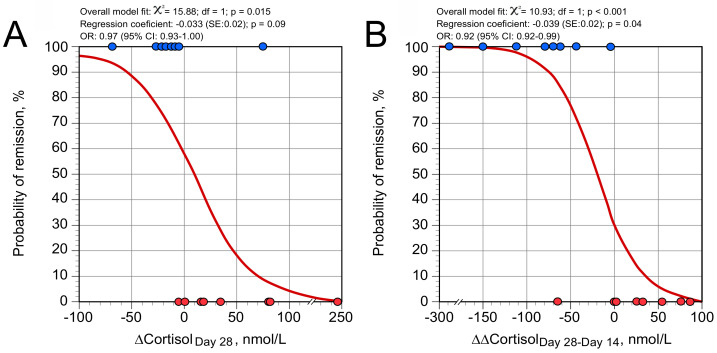
Predicted probability of remission/non-remission (on Day 42) based on (**A**) ∆COR_Day 28_ values after 4 weeks of antidepressant treatment (ADT), and (**B**) ∆∆COR_Day 28–Day 14_ (difference between ∆COR values after 4 and 2 weeks of ADT) using a logistic regression prediction curve. Logistic regression produces the regression coefficient (and standard errors (SEs) of estimate, and significance level) and odds ratio (OR, with 95% confidence interval) associated with each ∆COR _Day 28_ or ∆∆COR _Day 28–Day 14_ value.

**Table 1 brainsci-11-00425-t001:** Bio-clinical data on patients treated with either tianeptine or venlafaxine.

	Tianeptine	Venlafaxine	*p*
	(*n* = 8)	(*n* = 8)	
Age, years ^a^	35.6 ± 12.6	41.7 ± 7.6	0.34
Sex M/F, *n*	05-Mar	05-Mar	1
**HAM-D Scores**			
Day 0	25.9 ± 4.2	27.7 ± 5.0	0.59
Day 14	16.1 ± 4.6	14.5 ± 11.9	0.63
Day 28	11.5 ± 6.8	8.2 ± 6.9	0.39
Day 42	9.9 ± 9.1	8.2 ± 7.9	0.79
**Remitters**, *n*	4	4	1
**Apomorphine test**			
COR_BL_, nmol/L			
Day 0	200.1 ± 65.2	231.2 ± 78.2	1
Day 14	293.2 ± 108.1	227.2 ± 76.9	0.2
Day 28	296.4 ± 118.4	267.2 ± 98.2	0.72
∆COR, nmol/L			
Day 0	69.4 ± 127.1	95.2 ± 115.6	0.27
Day 14	30.5 ± 102.4	82.0 ±87.9	0.18
Day 28	−0.2 ± 17.3	47.2 ± 96.1	0.44
**Post-DST COR_max_**			
Day 0	49.1 ± 58.5	23.1 ± 10.0	0.95
Day 14	23.5 ± 9.2	28.6 ± 18.4	0.79
Day 28	23.0 ± 10.7	25.7 ± 15.3	0.91

^a^ Values are mean ± SD. HAM-D indicates the 17-item Hamilton Rating Scale for Depression; COR, cortisol; COR_BL_, basal cortisol; ∆COR, peak concentration minus basal concentration; DST, dexamethasone suppression test; COR_max_, highest post-DST COR level. Comparisons between groups were tested with Mann–Whitney two-tailed U test (quantitative variables) or Fisher’s exact test (qualitative variables).

**Table 2 brainsci-11-00425-t002:** Bio-clinical data of depressed groups defined by antidepressant treatment responses.

	Remitter Patients	Non-Remitter Patients	*p*
	(*n* = 8)	(*n* = 8)	
Age, years ^a^	39.4 ± 7.8	38.0 ± 13.3	0.95
Sex M/F, *n*	05-Mar	05-Mar	1
**HAM-D Scores**			
Day 0	27.0 ± 5.0	26.6 ± 4.5	0.91
Day 14	11.7 ± 5.7	21.9 ± 5.8	0.003
Day 28	4.7 ± 3.3	15.0 ± 5.4	0.001
Day 42	2.1 ± 2.2	16.0 ± 5.6	0.0008
**Venlafaxine/Tianeptine**	04-Apr	04-Apr	
**Apomorphine test**			
COR_BL_, nmol/L			
Day 0	233.1 ± 87.4	198.5 ± 78.2	0.6
Day 14	244.2 ± 75.0	276.2 ± 117.8	0.67
Day 28	266.0 ± 107.1	297.6 ± 110.0	0.57
∆COR, nmol/L			
Day 0	49.4 ± 84.4	115.2 ± 142.5	0.57
Day 14	81.6 ± 97.6	30.9 ± 87.9	0.14
Day 28	−10.5 ± 39.3	57.5 ± 81.5	0.005
∆∆COR, nmol/L			
Day 14–Day 0	32.2 ± 151.5	−84.4 ± 119.3	0.16
Day 28–Day 14	−92.1 ± 90.9	26.6 ± 48.4	0.002
**Post-DST COR_max_**			
Day 0	41.7 ± 52.6	30.5 ± 32.6	0.46
Day 14	20.7 ± 9.1	33.4 ± 17.1	0.31
Day 28	18.6 ± 6.4	30.1 ± 15.4	0.09

^a^ Values are mean ± SD. HAM-D indicates the 17-item Hamilton Rating Scale for Depression; COR, cortisol; COR_BL_, basal cortisol; ∆COR, peak concentration minus basal concentration; ∆∆COR, difference between ∆COR Day 14–Day 0 (or Day 28–Day 14); DST, dexamethasone suppression test; COR_max_, highest post-DST COR level.

## Data Availability

Data are available on request.

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
