# Peer review of "Neuroendocrine Assessment of Dopaminergic Function during Antidepressant Treatment in Major Depressed Patients"

_brainsci, 2021, doi:10.3390/brainsci11040425_

Round 1

Reviewer 1 Report

This manuscript reports findings from a study examining changes in HPA axis function (as an indirect indicator of DAergic neurotransmission) during the course of antidepressant treatment in a sample of depressed in patients (inpatient healthy controls were used as a baseline comparison). The results suggest that blunted ΔCOR after 4 weeks of antidepressant treatment were associated with a greater likelihood of clinical remission. Although it would be tempting to conclude that changes in HPA axis function following antidepressant treatment is what determines clinical effectiveness of the antidepressant, the conclusions of this study are more complex in that antidepressant treatment did not significantly alter ΔCOR. This suggests that links between week 4 ΔCOR and the likelihood of remission was likely due to factors other than the pharmacological effect of the medication.

The study addresses an interesting question, however the number of potential confounds present combined with the small sample and lack of a placebo control condition render their findings tenuous. Although the authors have highlighted the small sample size as a limitation and have mentioned a number of other limitations that relate to the study design, the overall conclusions have not been tempered in light of these significant limitations. For example, the conclusion of the abstract states “Our results suggest that following chronic ADT desensitization of postsynaptic DA receptors connected with the regulation of the HPA axis at the hypothalamic level is associated with clinical remission.” This conclusion is not supported by the data because (a) treatment with antidepressants did not change ΔCOR and therefore cannot be said to have caused “desensitization of postsynaptic DA receptors connected with the regulation of the HPA axis”, and (b) the lack of a placebo condition means that any effects of time observed on ΔCOR cannot be directly attributed to antidepressant effects.

In terms of additional confounds, I don’t think these can be adequately controlled for in this sample size. For example:

  1. Including two different classes of antidepressant medications with different mechanisms (i.e., n=8 per group) makes interpretation of the lack of changes in ΔCOR over time difficult to interpret. Is this due to variable effects of the different medications on ΔCOR in terms of dose-response relationship, timing of effects etc? The small sample size means these effects cannot be robustly examined.
  2. Although the authors note that correction for multiple comparisons was not applied because all analyses were determined a priori, there are several pairwise comparisons reported in the results that are not outlined in the statistical analysis plan, and which therefore should have had a correction applied. For example, the statistical analysis section implies that a regression model will be used to assess the relationship between ΔCOR and clinical remission, however in the results section this is supplemented with a series of uncorrected pairwise comparisons. This form of uncorrected multiple testing is especially problematic given the small sample.
  3. It looks like some of the participants were smokers. Nicotine has potent effects on the mesolimbic dopamine system and these effects are evident during both acute nicotine administration and during nicotine withdrawal. Therefore, it is likely that smokers who were required to abstain from smoking for several hours on the morning of the APO test may have begun to go into withdrawal, confounding measures of DAergic function at the time of testing.

Other more minor issues that should be addressed include:

  1. A rationale for using a DST – this is not described in the introduction and only first mentioned in the procedures section
  2. A description of who the healthy controls were. They are listed as being hospitalized but the inclusion/exclusion criteria imply that they did not have a psychiatric condition and were otherwise medically healthy. Why were they hospitalized?
  3. Links between ΔCOR values and HAM-D values should be shown in scatterplots. The correlations are quite high, and I would be interested to see if this relationship were being driven by one or two outliers.

Reviewer 2 Report

I have read the manuscript " Neuroednocrine assessment of dopaminergic function during antidepressant treatment in major depressed patients". In my opinion this paper is interesting, studu was well planned and described. The great value of the manuscript is the part of the depp described and concerned limitations connected with presented clinical study. Mabey same of these statements should be presented also in the abstract - than it will be clear for the readers that presented results and hypothesis of different dopaminergic function during antidepressent treatment need futher investigations. 

Reviewer 3 Report

The authors investigated the impact of cortisol response to dexamethasone suppression test as well apomorphine during antidepressant treatment with venlafaxine and tianeptine. The main finding of the study is that blunted cortisol response at four weeks after antidepressant treatment in the apomorphine test is different between remitters and non-remitters. I have several minor remarks that will hopefully improve this work:

  • please complete line 137, something is missing here.
  • I wonder why absolute COR values at baseline and day 14 are substantially different between remitters and non-remitters but not statistically significant (table 2)
  • Was there a placebo group available (fort cortisol/Apo tests)?
  • Was there a power analysis performed before including subjects? If not, why not?  
